# Optimization study of intelligent accounting manager system modules in adaptive behavioral pattern learning and simulation

Yifan Wang[1], Rongjie Qin[2] and Musadaq Mansoor[3]

[1] Shenzhen MSU-BIT University, Shenzhen, Guangdong, China
[2] Wuhan Technology and Business University, Wuhan, Hubei, China
[3] School of Computing Sciences, Pak-Austria Fachhochschule Institute of Applied Sciences and Technology, Haripur, Pakistan



## ABSTRACT

Within the ambit of the digital epoch, the advent of adaptive learning technologies heralds a paradigmatic shift in the realm of accounting management, garnering increasing scrutiny for augmenting learning outcomes *via* more sagacious educational methodologies and refining the accounting management protocols through the employment of sophisticated optimization techniques. This manuscript delineates an avant-garde health classification schema for accounting management, termed the A-CHMM-FD methodology, which amalgamates the merits of the Analytic Hierarchy Process (AHP) with the Coupled Hidden Markov Model (CHMM) to enhance the precision and efficacy of risk detection. Utilizing the AHP modality, we quantify diverse accounting metrics, subsequently subjected to independent scrutiny *via* the CHMM. This results in an exhaustive evaluation of entities as healthy, at-risk, or high-risk employing fuzzy delineations. Empirical validation on publicly available financial risk datasets and the pragmatic deployment of bespoke datasets affirm the superior efficiency and precision of the proposed framework. Applying this methodology within the health classification of accounting management emerges as efficacious, charting a novel technological trajectory for managing accounting risks and offering fresh perspectives on the nurturing of accounting understanding and the acquisition of knowledge.

## INTRODUCTION

Adaptive learning technologies reign supreme atop the hierarchy of educational advancements. Among the myriad strengths inherent in technology, its unparalleled capacity to facilitate the widespread dissemination of innovation stands resplendent, with adaptive learning technology serving as a quintessential exemplar. Crafting high-caliber adaptive learning systems necessitates the adept guidance of seasoned learning architects, who meticulously orchestrate the delineation of learning objectives into discrete parcels of knowledge and skill. Furthermore, they orchestrate the conception of a holistic learning journey that steers content creation on a grand scale (*Muñoz et al., 2022*). Hence, the

Corresponding author
Rongjie Qin, Qinrj@aliyun.com

imperative lies in harnessing the prowess of AI technology to engender an augmented *corpus* of content, thereby furnishing judicious evaluation of adaptive learning outcomes to enhance student efficacy. The realm of accounting management contends with a labyrinthine milieu characterized by dynamic regulatory shifts, market dynamics, and technological frontiers. Only through adaptive behavioral learning can we adeptly navigate these vicissitudes, perpetually refining management methodologies and strategies in tandem (*Osadcha et al., 2020*).

The digital metamorphosis within the realm of accounting denotes the transition from conventional article-based documentation and manual methodologies to a realm of digitalization and intelligence propelled by the advent of digital technology (*Bhimani, 2020*). Embracing this paradigm shift is imperative for accounting management and learning in the contemporary landscape characterized by ceaseless technological evolution. The traditional *modus operandi* of accounting management is delineated in Table 1.

Traditional accounting practices predominantly hinge upon manual data entry and processing, thereby inherently susceptible to data inaccuracies and time lags. Conversely, integrating digital technology facilitates swift and precise entry and processing of accounting data, markedly enhancing efficiency and accuracy. Furthermore, the advent of accounting digital transformation propels the modernization of accounting information management (*Igou et al., 2023*). Traditional methods rely heavily on cumbersome article documents and manual filing systems, prone to inefficiencies and susceptibility to loss or damage.

The digital transformation within the accounting domain encompasses a multifaceted process of evolution, encompassing technological innovation, procedural refinement, and organizational adaptation. By harnessing the capabilities of digital technology, significant enhancements and refinements have been ushered into the realms of work methodologies and business paradigms within the accounting sphere, furnishing enterprises with heightened efficiency and precision in financial management and decision-making support (*Aulia, 2020*). Consequently, the imperative lies in effectuating digital transformation through the deployment of machine learning and deep learning methodologies empowered by augmented computational prowess. Machine learning techniques such as support vector machines (SVM) and random forest offer potent avenues for fraud detection, financial risk assessment, and predictive modeling applications. SVM, for instance, proves adept at discerning irregular transactions, while random forest excels in constructing predictive models for financial fraud detection. Meanwhile, deep learning methodologies, exemplified by convolutional neural networks (CNN) and recurrent neural networks (RNN), find utility in tasks such as image recognition and text categorization. CNN facilitates the recognition of bill images, whereas RNN adeptly categorizes financial report texts and conducts sentiment analysis (*Cho et al., 2020*).

In response to the difficulties of contemporary digital accounting learning transformations, alongside the burgeoning needs of accounting management and associated financial intelligence, this article endeavors to furnish a comprehensive analysis of enterprise accounting management challenges, conferring the subsequent contributions:

**Table 1 The accounting management content.**

| Management range | Content |
| --- | --- |
| Cashier | Account management, payment settlement, automatic reconciliation |
| Budget management | Budget formulation, budget execution benchmarking, budget analysis |
| Tax management | Electronic invoices, scanning and automatic invoicing, tax reporting. |
| Financial reports | Operations, cost analysis, macro environment. |

(1) In response to the accounting management requisites, AHP is advocating for undertaking a quantitative analysis of multiple indicators, commencing from the four principal facets encapsulated within its management purview.

(2) The proposition entails completing an analysis of the robustness of corporate accounting management, employing a decision-level fusion framework denoted as A-CHMM-FD. This framework integrates AHP index quantification with CHMM and Fuzzy index decision-making methodologies.

(3) Empirical validation through public dataset scrutiny and practical application tests evinces commendable recognition accuracy across all obtained results, underscored by their inherent robustness and generalization performance.

The subsequent sections of the article are structured as follows: "Related Works" provides an overview of pertinent literature concerning risk management and accounting management. The proposed framework is delineated in "Methodology". "Experiment Result and Analysis" elucidates the experimental setup and presents the results. Finally, the article culminates with a discussion and conclusion.

## RELATED WORKS

### Smart management and risk analysis

With the perpetual advancement of computer technology and big data analytics, intelligent financial systems have emerged as indispensable tools for conducting nuanced risk analyses, streamlining accounting audits, and enhancing overall work efficiency. *Mann (2019)* contends that machine learning epitomizes the deployment of algorithms and statistical models imbued with a semblance of cognitive prowess. Particularly within the realm of accounting, it can amalgamate findings from financial and non-financial analyses, thereby furnishing decision-makers with invaluable support (*Mann, 2019*). *Li (2010)* posits that machine learning holds the potential to assess the predictability of future earnings, thereby facilitating budgetary and strategic financial decision-making for corporate management. *Perols (2011)*' study underscores machine learning's efficacy in detecting fraudulent activities within financial statements. *Ovaska-Few (2017)* asserts that AI technology can efficiently solve repetitive, mundane tasks by leveraging vast datasets and machine learning algorithms. For instance, it can expedite the evaluation of user data in a fraction of the time required for traditional audit sampling (*Ovaska-Few, 2017*).

Similarly, *Zhou (2020)* observes the widespread adoption of AI technology among auditors and accounting firms to detect fraudulent invoices, dramatically reducing

processing times from months to days. *Hazar (2019)*'s research showcases the utility of machine learning in modeling and forecasting within financial analysis, thereby facilitating business analysis applications. Min pioneers the integration of SVM models in corporate bankruptcy prediction, culminating in the development of a novel early warning model. Comparative assessments against multiple discriminant analysis, logistic regression, and BPNN models affirm the SVM model's superior prediction accuracy and generalization capabilities (*Min & Lee, 2005*). *Huang, Chen & Wang (2007)* advance the construction of a financial risk scoring model based on the SVM framework, demonstrating its heightened warning accuracy compared to alternative models such as BP neural networks and decision trees, particularly in scenarios with limited sample sizes.

## Current status of accounting management research

IFAC delineates management accounting as the comprehensive process encompassing the collection, analysis, recognition, measurement, presentation, interpretation, and dissemination of information management utilized for strategic planning, evaluation, and control. This facilitates the judicious utilization of resources and the assumption of managerial accountability (*Vysochan et al., 2021*). The American Institute of Management Accountants advocates for the integrated role of management accounting in shaping and executing organizational strategy, thereby serving as a linchpin in managerial decision-making processes. Meanwhile, the Chartered Institute of Management Accountants (CIMA) underscores the adherence to accounting and financial management principles, with management accounting spanning strategic management, operational planning and control, incentive strategy formulation, provision of decision support, financial and non-financial reporting, asset preservation, corporate governance, and internal control (*Hertati, Safkaur & Simanjuntak, 2020*). *Alsharari & Youssef (2017)* delineates the transformative trajectory of management accounting post-public sector reform, emphasizing the importance of comprehensive budgeting as a catalyst for top-down transformative change. This entails establishing a hierarchical system that furnishes pertinent information to all levels of management, facilitating strategic planning with comprehensive and timely insights across key accounting categories (*Alsharari & Youssef, 2017*). *Schuster, Heinemann & Cleary (2021)* elucidates management accounting as a principal avenue for augmenting organizational performance, encompassing methodologies and tools for cost management, management control, and broader managerial functions. Herbert advocates for adopting the shared service organization (SSO) model, positing it as a vehicle for consolidating internal support services while aligning with corporate objectives and enhancing internal management oversight. Through centralization and customer-centric approaches, SSOs yield cost efficiencies and service quality enhancements, thereby consolidating control and expertise within the corporate hierarchy (*Herbert & Seal, 2012*). *Yang (2014)* underscores the imperative for group companies to establish financial shared service centers, centralizing resource-intensive operations to unlock value creation potential while facilitating efficient processing and analysis of enterprise financial data. *Halbouni & Nour (2014)* espouses a perspective of management accounting innovation, attributing globalization, information technology, and enterprise-scale as catalysts for

transformation. Information technology emerges as a pivotal driver of management accounting innovation, necessitating investments in technology infrastructure and periodic updates to management accounting systems to optimize effectiveness. Moreover, training initiatives are crucial to augment the proficiency of management accountants in leveraging emerging technologies (*Halbouni & Nour, 2014*).

The aforementioned research underscores the pervasive focus on financial risk within the domain of accounting management, with modeling endeavors addressing various risk categories demonstrating considerable maturity. However, there remains ample scope for refining the selection of modeling methodologies. Presently, traditional machine learning techniques such as SVM and backpropagation neural networks (BPNN) dominate the landscape and have yielded promising results. Nevertheless, practical application reveals the inherent volatility of data and the imperative of continually refining models to align with evolving environmental dynamics. In light of these considerations, this article endeavors to delve deeper into the realm of risk identification within the context of accounting management, acknowledging the multifaceted nature of the discipline. Moreover, recognizing that accounting management comprises a multitude of factors, there arises a need for quantitative analysis across diverse indicators to facilitate comprehensive risk analysis and informed management decision-making.

# METHODOLOGY

## Risk index selection and quantification

In accordance with the factors outlined in the introduction pertaining to the accounting management process, this article employs the four dimensions delineated in Table 1 for modeling and quantifying the associated indicators. Notably, utilizing financial indicators as an exemplar, we undertake a quantitative analysis employing the AHP. Within the audit framework of financial management, it becomes imperative to quantify key monitoring indicators to bolster overall model performance. Considering the challenges inherent in the financial industry's learning curve and its distinctive characteristics, this article adopts a risk-source-oriented classification. Specifically, from the vantage point of risk origination, we delineate six perspectives: profitability, solvency, asset quality, operational capacity, liquidity, and developmental capacity. Subsequently, appropriate indicators are selected from each perspective to furnish quantitative insights, thereby offering a comprehensive depiction of prevailing financial risks with a succinct set of indicators.

As depicted in Fig. 1, the analysis of ubiquitous data within the regulatory framework of the accounting industry reveals six primary facets of financial management content. Utilizing the AHP, we refine these aspects to distill over 20 quantitative indicators, culminating in the construction of a model underpinned by artificial intelligence methodologies.

The severity of an indicator, which is proposed to be called health index in this study (*Lyu et al., 2020*). A part of the known data is selected as the training dataset, denoted as $A$; This part of the health degree index is denoted as $\hat{y}, \hat{y} \in (0, 1)$. The AHP was applied to study the quantitative representation of the health degree index. $\hat{y}$ AHP is applied to study the quantitative representation of the healthiness index. Initially, multiple experts assess

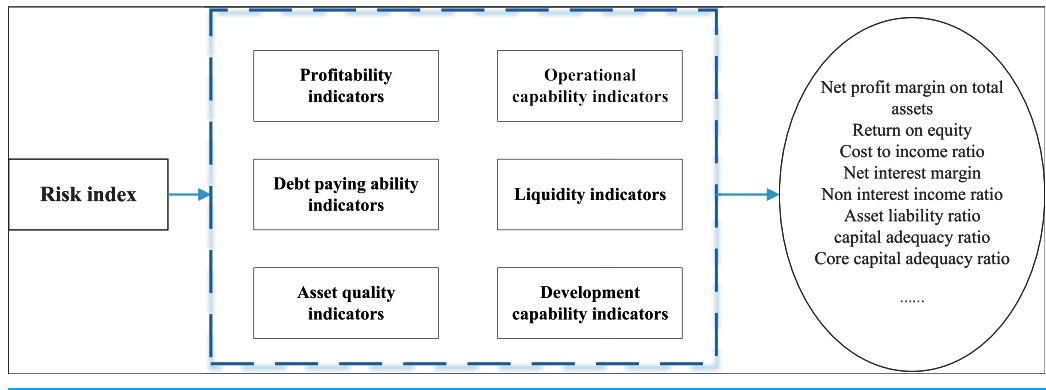

**Figure 1 The indicator for the risk analysis.**

enterprise performance subjectively, assigning scores within the range of 0 to 100. Subsequently, a judgment matrix is formulated based on the evaluations provided by these experts. Finally, evaluation weights are ascertained through the eigenvalues derived from the matrix. $W_t, (t = 1, 2, 3, 4, 5)$ Finally, the evaluation weights are determined based on the eigenvalues of the matrix. The evaluation weights are then mapped to the evaluation weights by the interval mapping function shown in Eq. (1). $W_t$ The evaluation weights are then mapped to the intervals as shown in Eq. $(0, 1)$ interval as shown in Eq. (1) to obtain the index of the patient's condition degree $\hat{y}$.

$$\hat{y} = \begin{cases} re - grade & \Delta W_i < \alpha \\ \lambda & \alpha \le \Delta W_i \le \beta \\ 0.2(i-1) + 0.2 * \Delta W_i & \Delta W_i > \beta \end{cases} \quad (1)$$

$$\lambda = \begin{cases} 0.2(i-1) + 0.2 * (\Delta W_i - \nabla W_j) & i > j \\ 0.2i - 0.2 * (\Delta W_i - \nabla W_j) & i < j \end{cases} \quad (2)$$

where $\Delta W_i$ is the largest weight value in $W_t, (t = 1, 2, 3, 4, 5)$ the value of the largest weight in, and $i(i \in \{1, 2, 3, 4, 5\})$ denotes the number of the degree of condition corresponding to the largest weight (see Table 1); the $\nabla W_j$ denotes $W_t, (t = 1, 2, 3, 4, 5)$ is the next largest weight in the list, and $j(j \in \{1, 2, 3, 4, 5\})$ denotes the number corresponding to the second largest weight; "re-grade" denotes that the clinical expert needs to re-score; "re-grade" is an empirical parameter; "re-grade" is an empirical parameter. $\alpha, \beta$ are empirical parameters. The quantification of the indicators for the four broad categories was completed with this background.

## CHMM-based feature evaluation for feature management metrics

Acknowledging the nuanced interplay between each quantitative indicator and the fluctuating nature of data across different time periods, this article adopts a time series processing approach for model development. Considering the manageable dimensionality of the data and the relatively moderate complexity of nonlinear relationships, as well as prioritizing interpretability in practical application, the hidden Markov model (HMM) method is chosen for model generation. Furthermore, to augment model performance and enhance the correlation among sub-indicators, the coupled hidden Markov model

(CHMM) method is employed for individual indicator grading. The CHMM model comprises two implicit state sequences and one visible state sequence, with one primary and one secondary implied state sequence (*Zou et al., 2022*).

As depicted in Fig. 2, the CHMM diverges from the conventional HMM by integrating multiple HMM chains, thereby enabling the state transition probabilities of the auxiliary state sequence to be influenced by the state of the primary state sequence (*Zhang et al., 2022*). The main advantage of CHMM over traditional HMM is that it dynamically adjusts state transitions and observation probabilities based on external features through a conditional dependency mechanism, making it suitable for modeling complex dynamic systems. In dynamic data experiments such as financial risk prediction, by introducing macroeconomic indicators and other conditional variables, CHMM shows higher flexibility and accuracy, and its predictive performance improves by 10–20% compared to traditional HMM in complex dynamic scenarios. This performance improvement mainly comes from CHMM's ability to better capture conditional dependencies and sudden changes in time series. In the realm of HMM-based methodologies, the fundamental concept revolves around effectuating state transitions and computing transfer probabilities between sequences. For CHMM, the primary state transfer probability can be expressed by Eq. (3):

$$P\left(q_t^M = s_j^M \mid q_{t-1}^M = s_i^M\right) = \omega(t)a_{ij}^M(t). \tag{3}$$

Auxiliary state transfer probabilities, affected by the main sequence of implied states: the

$$P\left(q_t^A = s_j^A \mid q_t^M = s_k^M, q_{t-1}^A = s_i^A\right) = \omega(t)a_{ij}^A(k,t). \tag{4}$$

To ensure the dynamic performance of the model, we introduce an adaptive weighting mechanism that allows the model to dynamically adjust its weights based on changes in data at each moment, thereby enhancing the flexibility and prediction accuracy of the model. This factor is calculated based on past observation data, time series volatility, and the need for risk assessment, in order to make corresponding adjustments to various parameters in the model. Obtained observation probabilities.

$$P\left(o_t \mid q_t^M = s_j^M\right) = b_j(o_t). \tag{5}$$

After completing the definition of the brother transfer matrix, the state initialization needs to be completed with the initial state probability distributions of the two HMM chains:.

$$\pi_i^M = P\left(q_1^M = s_i^M\right) \tag{6}$$

$$\pi_i^A = P\left(q_1^A = s_i^A\right). \tag{7}$$

State sequence probability.

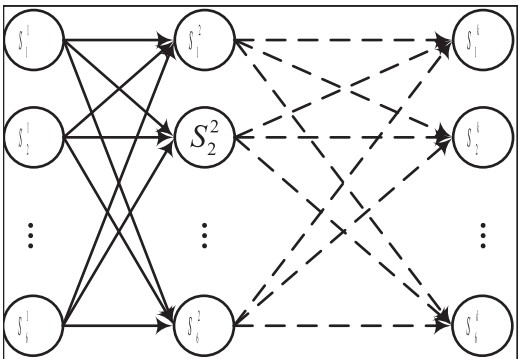

**Figure 2 The framework for the CHMM.** 

$$P\left(q_1^M, \ldots, q_T^M, q_1^A, \ldots, q_T^A\right) = \pi_{q_1^M}^M \cdot \pi_{q_1^A}^A \cdot \Pi_{t=2}^T a_{q_{t-1}^M q_t^M}(t) \cdot \Pi_{t=1}^T \Pi_{k=1}^N a_{q_{t-1}^A q_t^A}(k, t) \tag{8}$$

The observed sequence probability can then be expressed through Eq. (9):

$$P(o_1, o_2, \ldots, o_T) =$$

$$\sum_{q_1^M} \sum_{q_2^M} \cdots \sum_{q_T^M} \sum_{q_1^A} \sum_{q_2^A} \cdots \sum_{q_T^A} P\left(q_1^M, q_2^M, \ldots, q_T^M, q_1^A, q_2^A, \ldots q_T^A\right) \cdot \Pi_{t=1}^T b_{q_t^M}(o_t) \tag{9}$$

of which $q_t^M$ and $q_t^A$ denote the time $t$ the primary implied state and the secondary implied state, respectively. $o_t$ denotes the time $t$ observations of time, and $T$ denotes the length of the sequence. $a_{ij}^M(t)$ denotes the number of observations at time $t$ from the primary state $s_i^M$ transfer to the primary state $s_j^M$ the probability of the transition from the primary state to the primary state. $a_{ij}^A(k, t)$ Indicates the probability of moving from the primary state to the primary state at time $t$ from the auxiliary state $s_i^A$ transfer to the auxiliary state $s_j^A$. The probability of being affected by the primary state $s_k^M$ of the primary state. $b_j(o_t)$ Indicates the probability that a value is observed in the primary state $s_j^M$. The probability of observing the value $o_t$. The probability that the $\pi_i^M$ and $\pi_i^A$ denote, respectively, the probability that the system is in the primary state at time $t = 1$ the system is in the primary state $s_i^M$ and auxiliary state $s_i^A$ of the initial probability.

By leveraging the state transitions outlined above and analyzing pertinent data, we achieve the culmination of a unified risk identification model. Furthermore, in our pursuit of enhancing the overall efficacy of risk identification and ensuring a more precise assessment of the state of accounting management, we conduct additional analysis beyond the health risk assessment of individual features.

## Accounting management health assessment based on fuzzy index and CHMM features

To methodically and quantitatively assess the health of corporate accounting management across four dimensions, this study introduces a novel comprehensive evaluation method grounded in a fuzzy index model. This approach aims to analyze and evaluate the current state of corporate accounting management with precision and rigor (*Wu & Hu, 2020*). For this article, the health status of the company's accounting management is divided into

three levels, namely healthy, risky and high-risk, so that its set is denoted as $A = \{A_1, A_2, A_3\}$. The set of the above observation indicator factors is defined as $z = \{z_1, z_2, z_3\}$, and the weights of the observation indicator factors are defined as $A_{j(i)}, \ (j = 1, 2, 3; \ i = 1, 2, 3)$. The quantitative levels of the observables are defined as Level I, Level II, and Level III. The expression of the function of the observational indicator which can be differentiated into light and heavy is:

$$\alpha(z_i) = \begin{cases} I & 3 \ points \\ II & 2 \ points \\ III & 1 \ point \end{cases} \quad i = (1, \ 2, \ 3). \tag{10}$$

The factors of each observation indicator are quantified by the corresponding score, so the expression of judgmental affiliation function can be set as:

$$\begin{aligned} J_j &= \frac{A_{j(1)}\alpha(z_1) + A_{j(2)}\alpha(z_2) + A_{j(3)}\alpha(z_3)+}{A_{j(1)} + A_{j(2)} + A_{j(3)}} \\ &= \sum_{i=1}^{3} \frac{A_{j(i)}\alpha(z_i)}{\sum_{i=1}^{4} A_{j(i)}}. \end{aligned} \tag{11}$$

Upon finalizing the construction of the affiliation function and conducting the health degree classification assessment, we proceeded to develop the A-CHMM-FD framework for the intelligent financial management method, as illustrated in Fig. 3.

In analyzing accounting management content, our methodology unfolds in several stages. Initially, we scrutinize data pertaining to the four principal management contents delineated in Table 1, effectuating the quantification of multiple indicators through the AHP method. Subsequently, we leverage the CHMM method to independently analyze the four categories of indicators, thereby achieving assessments of health, risk, and high risk. Building upon this analysis, we undertake accounting management health classification utilizing the fuzzy index method, drawing insights from relevant data. The robustness of our model and the accuracy of risk identification are enhanced through the seamless integration of these two tiers. Moreover, we establish health score thresholds wherein scores exceeding 0.6 denote a healthy status, scores falling within the range of 0.4 to 0.6 indicate a risky status, and scores of 0.4 or below signify a high-risk status. These thresholds are substantiated through extensive research and validation. For the above risk assessment classification, we have divided it based on existing enterprise data in this article, and conducted statistical analysis according to the scoring effects of different enterprises in the local area. The results are shown in Fig. 4.

## EXPERIMENT RESULT AND ANALYSIS

### Dataset and experiment result

In consideration of the extensive array of data entailed in accounting management, direct access to comprehensive company data is often challenging at this stage. Thus, this article draws upon prior research on accounting management and financial risk (*Soin & Collier,*
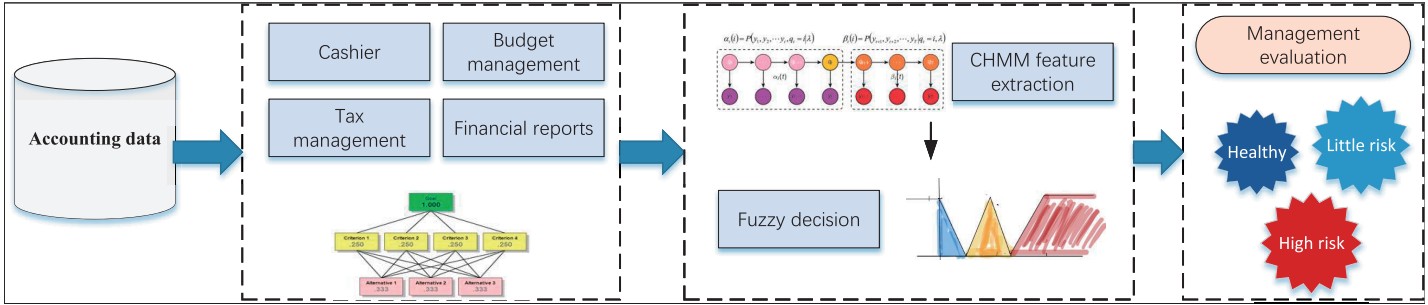

**Figure 3 The framework for the accounting health evaluation.**

*2013*), opting to utilize the S&P 500 as the public dataset for model analysis (*Alareeni & Hamdan, 2020*). This dataset encompasses 500 prominent listed firms in the US market, offering historical data on various parameters such as daily closing prices, volume, and volatility. By scrutinizing the historical data of the S&P 500, one can conduct stock market trend prediction and risk assessment. To facilitate model analysis and comparison within the realm of accounting management, we juxtapose risk predictions based on classical algorithms. The principal methods examined include HMM, SVM, BPNN, RNN, and long short-term memory (LSTM), alongside computation time considerations. In addition, considering the low number of feature dimensions, we did not use PCA or other methods for feature selection in this study. Instead, we tried to ensure the availability and interpretability of all features as much as possible for further analysis in the future.

During the model training process, we randomly divided the dataset into training and testing sets in a 5:1 ratio. At the same time, considering the parameter setting issues in CHMM and FD, we performed parameter initialization. Considering the partitioning task presented in this article, we set the hidden state of CHMM to 3, the learning rate to the most basic 0.01, and the maximum number of training iterations to 200.

Given the financial stability attributes of listed companies, during the testing phase with the public dataset, we focus solely on classifying companies into risky and healthy categories. The obtained identification accuracy results are illustrated in Tables 2 and 3, Figs. 5 and 6.

The results depicted in Fig. 4 underscore the notable advantage of the proposed method, particularly in the realm of high-risk identification, where it exhibits a precision of 0.913. This precision surpasses even that of single deep network methods such as RNN and LSTM. Additionally, the proposed method demonstrates a more balanced performance across the overall model, characterized by a more consistent distribution of its F1-score. Building upon these findings, we proceed to conduct health management identification.

In the health management recognition process, the A-CHMM-FD method proposed continues to outperform traditional machine learning methods and single deep learning networks. Across the three key metrics of precision, recall, and F1-score, the A-CHMM-FD method yields superior results, underscoring its enhanced health evaluation capability. This enhancement can be attributed to the effective integration of coupled hidden Markov

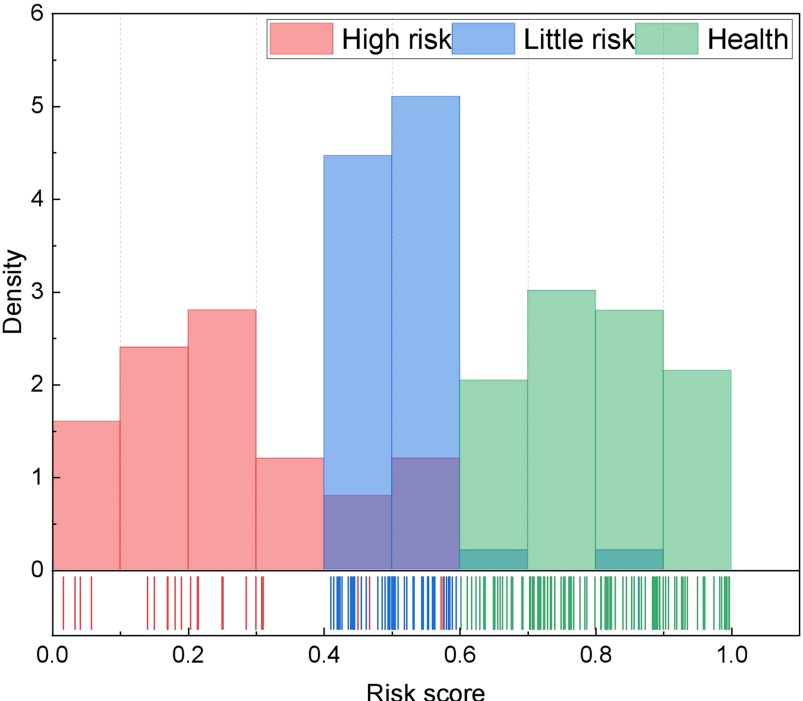

**Figure 4  The risk score distribution for the risk segmentation.**

and fuzzy indexes. Moreover, to evaluate the model's generalization performance and practical applicability, we conducted an analysis of the model's running time, as depicted in Fig. 7.

The analysis of running time reveals that while the model proposed exhibits slightly longer running times compared to traditional HMM and SVM methods, its overall recognition accuracy surpasses these methods. Moreover, despite achieving similar recognition accuracy to RNN and LSTM methods, the proposed model boasts lower actual running times than these two methods. This bodes well for the future deployment and application of the model, as it offers a favorable balance between computational efficiency and recognition accuracy.

## The practical test for the proposed framework

In our model testing, we conduct a comprehensive analysis and discussion on the precision, recall, and F1-score metrics, observing that the A-CHMM-FD method proposed exhibits superior robustness and generalization performance. Given the significance of accuracy in practical application, we prioritize a detailed discussion on this aspect. The specific results of this analysis are elucidated in Fig. 8.

In practical application, we present the recognition results for high-risk, low-risk, and healthy categories, respectively. Upon analyzing the accuracy across these three categories, we observe that the proposed method achieves recognition accuracies of 0.872, 0.859, and 0.855, respectively. These accuracies notably surpass those attained by HMM and SVM methods. Furthermore, the proposed method's overall results closely align with the

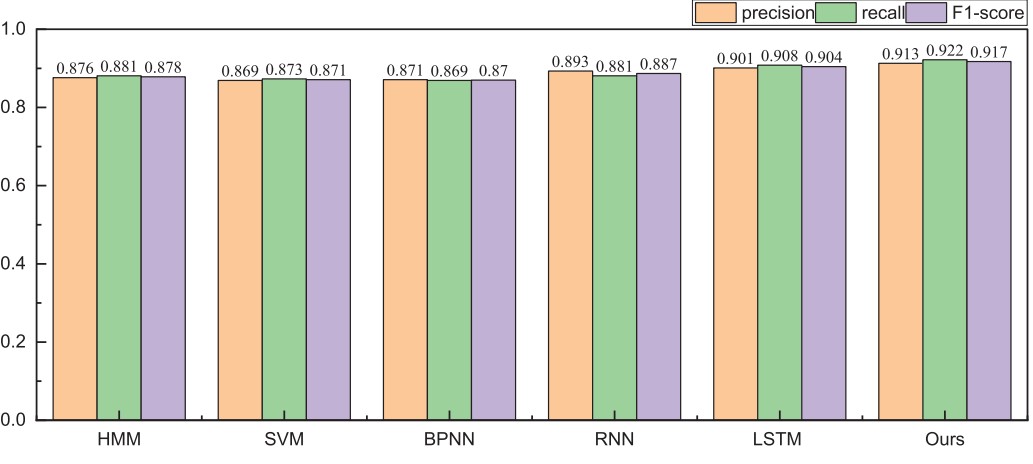

**Figure 5** The recognition result for the high risk.

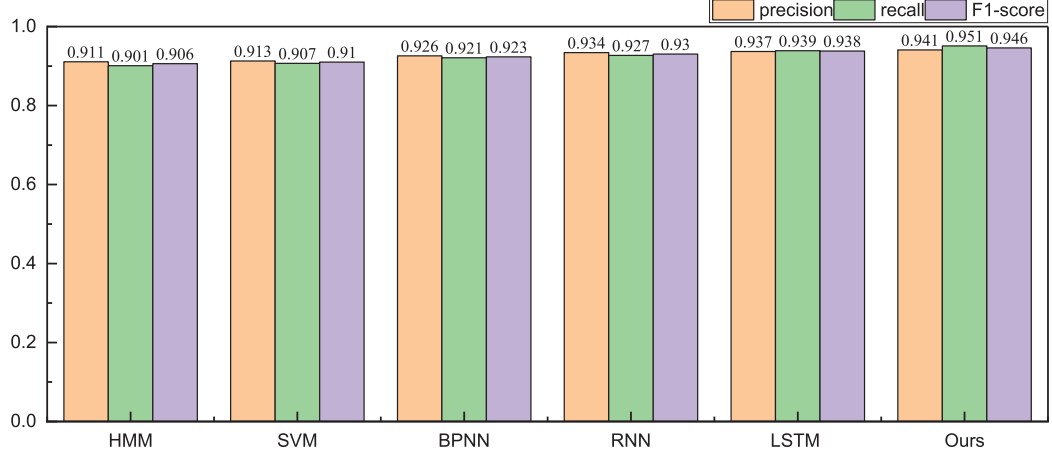

**Figure 6** The recognition result for the healthy management.

**Table 2** The recognition result for the high risk.

| Method | Precision | Recall | F1-score |
| --- | --- | --- | --- |
| HMM | 0.876 | 0.881 | 0.879 |
| SVM | 0.869 | 0.873 | 0.871 |
| BPNN | 0.871 | 0.869 | 0.870 |
| RNN | 0.893 | 0.881 | 0.887 |
| LSTM | 0.901 | 0.908 | 0.904 |
| Ours | 0.913 | 0.922 | 0.917 |

accuracies achieved by RNN and LSTM methods, while exhibiting greater efficiency in terms of runtime. In the field of high-risk object recognition, which is a key focus, the method proposed in this article has a recognition ability that is 2% higher than the widely used deep learning methods, which is very important for accounting management review.

**Table 3 The recognition result for healthy management.**

| Method | Precision | Recall | F1-score |
|---|---|---|---|
| HMM | 0.911 | 0.901 | 0.906 |
| SVM | 0.913 | 0.907 | 0.910 |
| BPNN | 0.926 | 0.921 | 0.923 |
| RNN | 0.934 | 0.927 | 0.9307 |
| LSTM | 0.937 | 0.939 | 0.938 |
| Ours | 0.941 | 0.951 | 0.946 |

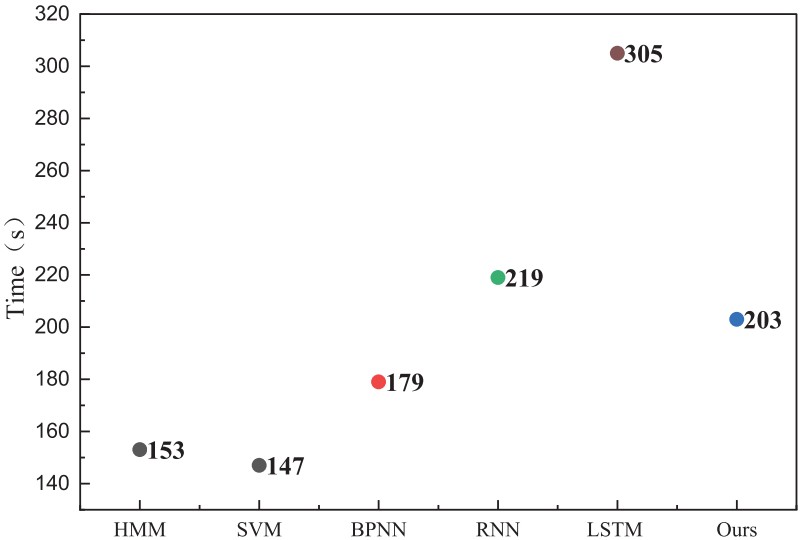

**Figure 7 The running time for the method comparison.**

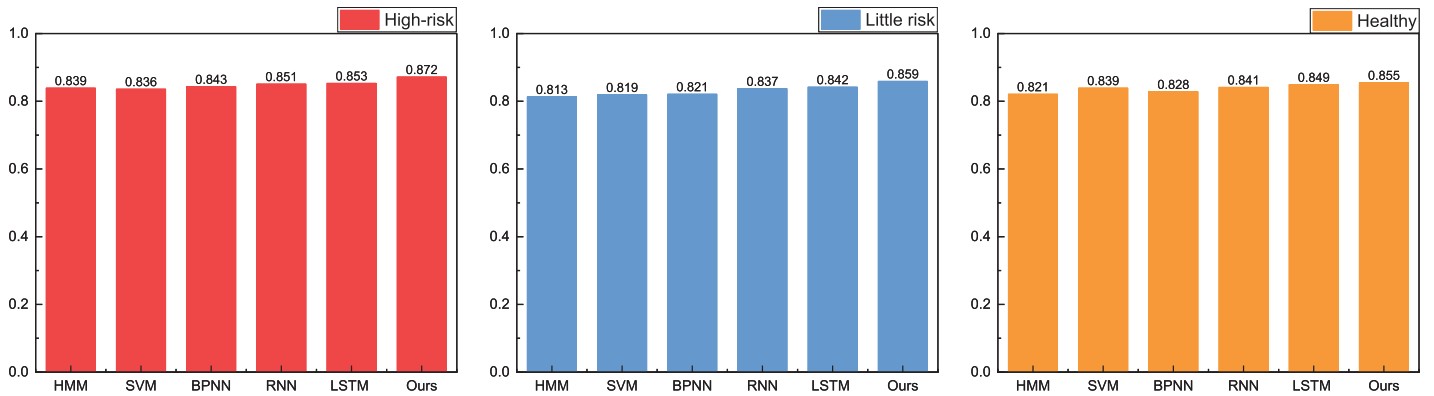

**Figure 8 The accuracy for the method comparison in the practical test.**

To provide a comprehensive analysis of the model's practical application performance, we select regional enterprises that meet the model test requirements. These enterprises are analyzed through a combination of manual annotation and machine learning method comparison. The results of this analysis are depicted in Fig. 9.

Figure 8 demonstrates that the recognition framework proposed achieves notably high recognition accuracy. Notably, in the artificial comparison of three risk levels, the framework's ability to recognize risks lower than LITTLE RISK significantly outperforms the manual labeling method. This highlights the framework's heightened sensitivity in identifying minor risks. Additionally, in the sample identification of high risk, the framework's overall performance slightly surpasses manual labeling, suggesting its potential to provide valuable insights for accounting management in future processes.

## Ablation experiment for the proposed framework

After comparing the model with the manual labeling method, we proceeded to conduct further ablation experiments to assess the effectiveness of different modules. Utilizing leave-one-out validation of the constructed models, we meticulously tabulated the results and compared the recognition effects of accounting managers before and after learning. The corresponding experimental findings are presented in Fig. 10.

In Fig. 10, the A-HMM and A-CHMM methods are mainly discussed, and statistics of high-risk and healthy management states are completed. It is not difficult to see from the results that the overall performance of the model has been improved after adding the fuzzy index module, and the coupling part in CHMM can improve the robustness of the model and achieve better recognition results. For the recognition results of high-risk and healthy objects shown in the figure, it can be seen that when using fuzzy decision-making for final classification, the overall performance is improved by about 3% due to the overlapping parts and corresponding data parts in fuzzy analysis. It can be seen that using fuzzy decision-making can to some extent solve the unbalanced performance of directly using softmax classification layers, and can ensure robustness to a certain extent. Through the verification process of different demand risk detection, it can be found that the A-CHMM-FD method proposed in this article has better risk identification performance.

## DISCUSSION

The A-CHMM-FD method proposed presents significant advantages in the realms of financial risk prediction and accounting management health classification. Firstly, by leveraging the comprehensive utilization of the AHP, multiple indicators are quantified and comprehensively assessed, enabling a multidimensional evaluation that enhances the comprehensiveness and accuracy of assessment outcomes. Secondly, the A-CHMM-FD method employs a multilevel model design that integrates AHP and CHMM methods, thereby bolstering the robustness and accuracy of the model. This design facilitates a nuanced understanding of the intricate relationships within the data, thereby enhancing the predictive capability of the model. Moreover, the application of the fuzzy index method for accounting management health classification adeptly addresses uncertainty and ambiguity in the data, thereby enhancing the model's applicability and robustness.

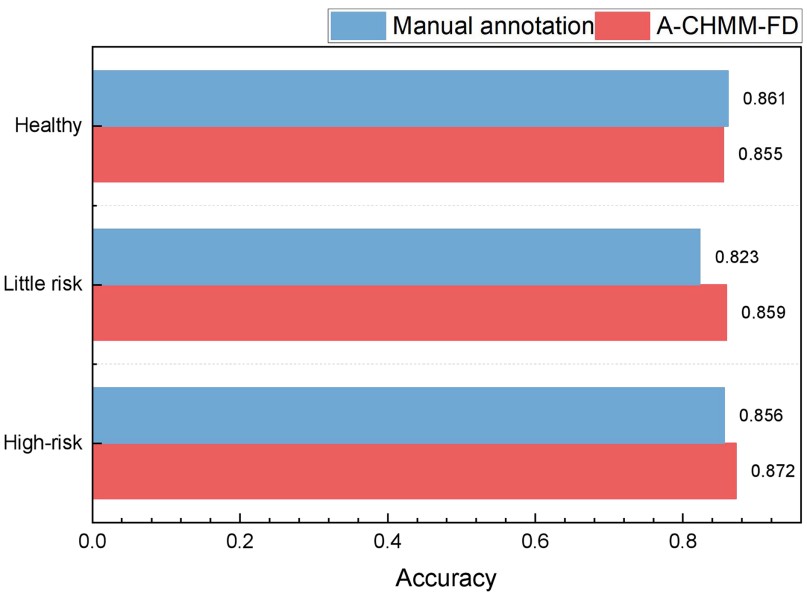

**Figure 9 The accounting risk recognition comparison in the practical test.**

Compared to methods such as HMM, SVM, BPNN, RNN, and LSTM, the A-CHMM-FD method demonstrates greater flexibility in handling uncertainty and ambiguity in financial data, thereby improving the model's applicability and robustness. While the overall recognition rate of A-CHMM-FD approaches that of deep learning methods in terms of computation time, its overall computation time and model interpretability significantly outperform deep learning methods. Despite marginally longer computation times compared to single HMM, SVM, and other traditional machine learning methods, the advantages of A-CHMM-FD are apparent. For the risk identification research conducted in this article, there are currently many model frameworks for analysis. However, this article hopes to provide a more reliable framework for local data through A-CHMM-FD to ensure the identification effect. Meanwhile, it provides reference for highly interpretable algorithms based on machine learning and traditional methods.

The intelligent accounting management assessment proposed holds significant promise for shaping the future trajectory of the accounting industry. Firstly, it stands to enhance decision-making efficiency and accuracy by leveraging advanced technologies and methodologies. Through more precise assessments of financial health and risk levels, accountants and financial managers can expedite decision-making processes while ensuring their accuracy. Secondly, intelligent accounting management assessment can mitigate risks and bolster resilience by timely identifying and predicting potential risks faced by enterprises, thereby providing corresponding countermeasures. Proactively managing risks in this manner helps reduce potential losses and enhances enterprises' ability to withstand risks. Moreover, intelligent accounting management assessment facilitates the digital transformation of the accounting industry, necessitating the adoption of advanced technologies such as big data and artificial intelligence. This shift towards

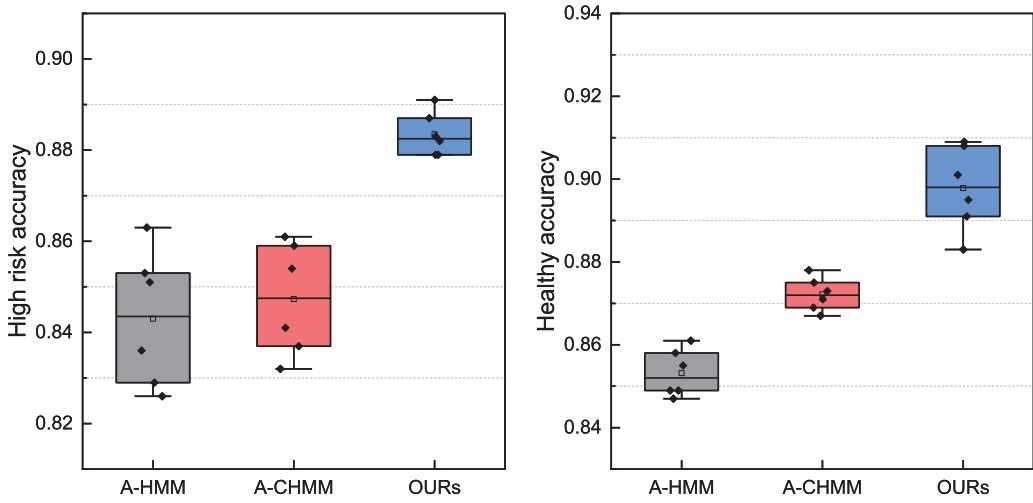

**Figure 10 The ablation experiment for the high-risk and healthy accounting recognition.**

digitalization and intelligence propels the industry towards greater efficiency and quality of work. However, in the process of future application, several key aspects demand special attention. Firstly, ensuring data quality and privacy protection is paramount. Robust measures must be in place to guarantee the accuracy and integrity of data, while strictly safeguarding customers' private information in compliance with relevant laws, regulations, and industry standards. Secondly, careful consideration should be given to technology application and AI ethics. Ethical and moral concerns surrounding AI must be addressed to ensure legal compliance and prevent adverse impacts on individuals or society. Lastly, talent training and technology updating are crucial. Intelligent accounting management assessment necessitates skilled professionals proficient in data analysis and artificial intelligence. Thus, investing in talent development and continuous technology updates is essential to meet the evolving demands of the industry. In conclusion, as adaptive learning and talent training through artificial intelligence methods continue to evolve, they play pivotal roles in shaping the future of the entire accounting industry.

## CONCLUSION

In the digital era, the integration of adaptive learning techniques and intelligent optimization methods marks a groundbreaking advancement in the realm of accounting management, particularly in risk identification and management. Through the proposed A-CHMM-FD framework, this study showcases the fusion of the AHP and CHMM to elevate the accuracy and efficiency of risk identification in accounting management. Experimental findings underscore the efficacy of the framework, revealing its capability for high-precision identification in real-world applications across both public datasets and self-constructed datasets. Notably, the framework outperforms traditional methods in terms of efficiency while maintaining comparable recognition precision to deep learning methods. Specifically, in risk tests utilizing public datasets, the framework achieves recognition precisions exceeding 90% in both scenarios, aligning with the performance of

deep learning methods. Furthermore, comparisons with manually labeled tests demonstrate consistent and promising results. Consequently, the A-CHMM-FD method paves a novel technical pathway for accounting risk management and heralds new vistas for the development of accounting talents and the acquisition of professional knowledge.

Despite the notable achievements of this study, several limitations and challenges persist in the domain of intelligent optimization and risk identification for accounting management. These challenges encompass the diversity and complexity of data, which may impede the generalization ability and accuracy of models. Future research endeavors could prioritize expanding and diversifying accounting management-related datasets, alongside refining and innovating model structures and algorithms to bolster the effectiveness and utility of risk identification. Furthermore, exploring deeper integration of adaptive learning techniques into the accounting management process to address evolving business environments and risk landscapes will be a pivotal avenue for future inquiry. By addressing these challenges and advancing research in these directions, the field of intelligent optimization and risk identification in accounting management can continue to evolve and meet the evolving needs of modern businesses and industries.

## ACKNOWLEDGEMENTS

The authors would like to thank the anonymous reviewers for their valuable comments on this article.

### Funding

The authors received no funding for this work.

### Competing Interests

The authors declare that they have no competing interests.

### Author Contributions

- Yifan Wang conceived and designed the experiments, performed the experiments, analyzed the data, performed the computation work, prepared figures and/or tables, authored or reviewed drafts of the article, and approved the final draft.
- Rongjie Qin conceived and designed the experiments, performed the experiments, analyzed the data, performed the computation work, prepared figures and/or tables, authored or reviewed drafts of the article, and approved the final draft.
- Musadaq Mansoor conceived and designed the experiments, analyzed the data, performed the computation work, prepared figures and/or tables, and approved the final draft.

### Data Availability

The S&P 500 data is available at Zenodo: Nitiraj Kulkarni, & Jagadish Tawade. (2024). Dataset: Global X S&P 500 Catholic Values ETF (CATH) Stock Performance [Data set]. Zenodo. https://doi.org/10.5281/zenodo.12554520.

## Supplemental Information

Supplemental information for this article can be found online at http://dx.doi.org/10.7717/peerj-cs.2684#supplemental-information.

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
