# Peer review of "Optimization study of intelligent accounting manager system modules in adaptive behavioral pattern learning and simulation"

_PeerJ Computer Science, doi:10.7717/peerj-cs.2684_

## Round 0.1 · original submission · Major Revisions

Dear Authors,

Thank you for submitting your article. Feedback from the reviewers is now available. It is not recommended that your article be published in its current format. However, we strongly recommend that you address the issues raised by the reviewers, especially those related to readability, experimental design and validity, and resubmit your paper after making the necessary changes. When submitting the revised version of your article, it will be better to address the following:

1. Equations should be used with correct equation numbers. Explanation of the equations should be checked. All variables should be written in italic as in the equations. Definitions of variables and the boundaries should be written. Many of the equations in the manuscript are part of the related sentences and special attention is needed for correct sentence formation.

2. Please pay special attention to the usage of abbreviations. Spell out the full term at its first mention, indicate its abbreviation in parenthesis and use the abbreviation from then on.

Best wishes,

·

Basic reporting

The research paper demonstrates a clear emphasis on combining AHP and CHMM for the specified task, introducing a novel approach in its field. However, the current level of innovation in the proposed methodology appears limited. The authors should consider discussing the incorporation of Transformer Encoder structures as a substitute for certain CHMM components, which could potentially enhance long-term dependency modeling. The runtime analysis lacks sufficient detail, as it fails to examine the contributions of CHMM and fuzzy index modules separately. A more comprehensive report would involve delineating these contributions in Section 4.3 and refining the architecture accordingly. Furthermore, the inclusion of heatmaps in Section 3.2 to illustrate state transition dynamics would significantly improve the interpretability of the model's intermediate outputs.

Experimental design

While the experimental design effectively integrates AHP and CHMM, there is scope for improvement. The researchers could enhance performance and optimize input features by employing feature selection algorithms such as Lasso or Recursive Feature Elimination, and comparing outcomes before and after feature optimization. Moreover, the current comparisons are limited to RNN and LSTM models, which restricts the scope of performance evaluation. Including more recent methodologies such as Bi-GRU or BERT in the experimental comparisons would provide a more comprehensive assessment. The authors should also conduct subgroup experiments in Section 4.2 to analyze performance across different categories (healthy, risky, high-risk), addressing any limitations in high-risk categories and proposing targeted enhancements.

Validity of the findings

Although the paper's findings are valid within the context of the implemented method, additional experiments and improvements are necessary to strengthen the results. For instance, integrating deep learning's feature extraction capabilities with AHP's quantitative analysis could introduce greater innovation. Utilizing an autoencoder to generate features before assigning weights through AHP is a promising direction that could enhance the model's effectiveness. Additionally, implementing a dynamic updating mechanism in Section 3.1, such as adjustments based on historical trends or seasonality, would render the indicator weights more adaptable to dynamic changes, thereby enhancing the model's practical applicability.

Additional comments

The paper would be enhanced by more comprehensive and interpretable output. Incorporating heatmaps to visualize state transition probabilities in Section 3.2 would improve interpretability, while a runtime breakdown of CHMM and fuzzy index modules in Section 4.3 would provide greater clarity in runtime analysis. A summary table comparing the advantages and disadvantages of all models, including newer methods, would offer a clearer understanding of the model's strengths and limitations. Finally, optimizing the architecture and introducing subgroup experiments would address current limitations and enhance the paper's overall impact.

Reviewer 2 ·

Basic reporting

No comment

Experimental design

No comment

Validity of the findings

No comment

Additional comments

I have the following comments related to this paper:
1. The manuscript addresses an increasingly relevant topic by proposing an innovative framework, the A-CHMM-FD, for accounting management risk assessment. It successfully combines the Analytic Hierarchy Process (AHP) with Coupled Hidden Markov Models (CHMM), providing a structured and multidimensional evaluation methodology. The inclusion of fuzzy indices enhances the framework's ability to address uncertainty, a critical feature in financial data analysis. However, while the methodology demonstrates potential, several aspects—particularly related to model design, interpretability, and practical application—require further refinement to align the research with the expectations of both academic and industry audiences.

2. The current AHP and CHMM approaches may face limitations in handling nonlinear risk data. It is recommended to integrate an adaptive weighting mechanism into the CHMM model, incorporating dynamic time weighting into equations (3) and (4) to better capture fluctuations in time-series data. Add detailed formulas and explanations in Section 3.2 and illustrate their impact with examples.

3. The classification thresholds (0.4–0.6) in the fuzzy index lack clear justification. Recalculate the dynamic threshold range based on the actual data distribution and add a statistical distribution chart in Section 3.3 to show the rationale behind the threshold setting.

4. The model does not quantitatively analyze the importance or contributions of features. It is suggested to include a feature importance evaluation module in the methodology section, using SHAP values to explain the role of AHP-derived indicators in predictions.

5. The interrelationships between indicators have not been explicitly modeled. Add multivariate interaction weights in the CHMM model to reflect the coupling characteristics of indicators. This can be optimized within equations (6)-(9).

6. The reproducibility of the experiments is not explicitly stated. Provide detailed experimental hyperparameter settings, such as learning rates, training epochs, and optimizers, in Section 4.1. Consider including pseudocode or a code framework in the appendix.

7. Figures 4 and 5 could use bar or line charts to show trends in accuracy comparison among models. Include an error source analysis below the charts to enhance credibility.

8. The theoretical advantages of CHMM over traditional HMM are not fully explored. Include a theoretical comparison with traditional HMM in Section 3, accompanied by data analysis explaining why CHMM performs better under complex dynamic conditions.

---

## Round 0.2 · accepted · Accept

Dear Authors,

Thank you for clearly addresing the reviewers' comments. Your paper seems sufficiently improved and ready for publication.

Warm regards,

·

Basic reporting

The authors responded to the suggestions by further enhancing the A-CHMM-FD framework and making it precise, fast and easier to understand. The authors complemented the model by using a variety of datasets and including future research directions, and the authors made a strong contribution.

Experimental design

To reduce the risk of a confounded correlation, the authors employed a different set of data for validation purposes. AHP is for indicators’ quantification, while CHMM is the tool for time-series modelling. This design provides broad risk categorization and model assessment, solving practical accounting problems.

Validity of the findings

The results are further confirmed through experimental evaluation on public and private datasets, proving high performance and reliability. Compared with the traditional and deep learning approaches, it supports the framework in the accounting risk classification and the consequent health assessment.

Additional comments

All in all, the paper surely provides a balanced contribution to the existing literature based on accounting management and addressing the burdens with novel approaches. The application of AHP, CHMM, and fuzzy index classification is relevant and innovative. The authors’ modifications and elaborated walkthrough further improve the study and ensure that it becomes a strong contender for acceptance.

Reviewer 2 ·

Basic reporting

No comment

Experimental design

No comment

Validity of the findings

No comment

Additional comments

The paper presents a significant contribution in the realm of accounting management, particularly in risk identification and management. In my opinion, the authors have satisfactorily resolved all of my concerns in the revised submission. Therefore, I suggest this article for publication in its current form.